# Analyzing and Resolving Infeasibility in Flux Balance Analysis of Metabolic Networks

**DOI:** 10.3390/metabo12070585

**Published:** 2022-06-23

**Authors:** Steffen Klamt, Axel von Kamp

**Affiliations:** Analysis and Redesign of Biological Networks, Max Planck Institute for Dynamics of Complex Technical Systems, Sandtorstr. 1, 39106 Magdeburg, Germany; vonkamp@mpi-magdeburg.mpg.de

**Keywords:** constraint-based modeling, metabolic flux analysis, mass balances, weighted least-squares, quadratic programming, *Escherichia coli*

## Abstract

Flux balance analysis (FBA) is a key method for the constraint-based analysis of metabolic networks. A technical problem may occur in FBA when known (e.g., measured) fluxes of certain reactions are integrated into an FBA scenario rendering the underlying linear program (LP) infeasible, for example, due to inconsistencies between some of the measured fluxes causing a violation of the steady-state or other constraints. Here, we present and compare two methods, one based on an LP and one on a quadratic program (QP), to find minimal corrections for the given flux values so that the FBA problem becomes feasible. We provide a general guide on how to treat infeasible FBA systems in practice and discuss relevant examples of potentially infeasible scenarios in core and genome-scale metabolic models. Finally, we also highlight and clarify the relationships to classical metabolic flux analysis, where solely algebraic approaches are used to compute unknown metabolic rates from measured fluxes and to balance infeasible flux scenarios.

## 1. Introduction

Various mathematical modeling techniques have been developed to facilitate the computer-aided analysis and exploration of complex metabolic networks. Constraint-based modeling is a particular class of those methods that is based on the network structure (stoichiometry), mass balances of metabolite concentrations in a steady state and certain physicochemical (thermodynamics, reversibility) and biological (e.g., maximal total amount of enzymes) constraints [1]. Within this setting, constraint-based modeling techniques analyze stationary flux distributions to uncover important properties of the metabolic network or to predict metabolic phenotypes [2]. A key method in this field is flux balance analysis (FBA), which searches for optimal metabolic flux distributions maximizing (or minimizing) a certain objective function, such as maximization of growth [3,4]. 

One of the oldest applications of constraint-based modeling is metabolic flux analysis (MFA), where measurements of certain reaction rates are used to derive unknown reaction rates in a given metabolic network [5,6,7,8]. Measurements typically comprise the growth rate and exchange rates of substrates and products, but some reaction rates may also be known from biological knowledge (e.g., inactive reactions under certain conditions) or might be fixed due to certain assumptions. Herein, we do not consider ^13^C-MFA, which uses measurements of isotopic labeling patterns from ^13^C-tracer experiments to resolve unknown metabolic rates [8]. 

The steady-state condition together with the known reaction rates lead to a system of linear equations, which, in classical MFA, is analyzed and solved by methods from linear algebra [7,9,10,11]. Depending on the network structure and the selection of reactions with measured or fixed values, there may be redundancies in the known rates that can lead to inconsistencies, i.e., there is no feasible flux distribution in the network satisfying the known fluxes and the steady-state conditions. Those inconsistencies have been resolved and treated with the least-squares approaches [7]. However, in contrast to certain methods, such as FBA, classical MFA does not consider any further constraints. In particular, it cannot deal with inequalities specifying, for example, reaction reversibilities, lower or upper bounds of reaction rates, or other more global constraints, such as limitations on enzyme abundances. On the other hand, while FBA can easily integrate these constraints, a systematic approach to properly handle and resolve inconsistencies in FBA scenarios, especially in the context of measured reaction rates, has, to the best of our knowledge, not been presented to date. In this work we introduce generic approaches to treat infeasibility in metabolic network models with arbitrary linear constraints. These approaches are based on linear or non-linear (quadratic) programs and seek to introduce minimal corrections to make an infeasible system feasible. We also clarify the relationships of this generalized (FBA-based) MFA with classical MFA, which relies solely on algebraic calculations. We also discuss practical examples of how infeasible FBA scenarios can arise in core and genome-scale models of *Escherichia coli*. 

## 2. Preliminaries

### 2.1. Definitions

We consider a metabolic network with m metabolites and n reactions represented by a stoichiometric matrix N∈ℝm×n. A fundamental assumption for all constraint-based modeling techniques is that the metabolism is in a steady state, i.e., the concentrations of internal metabolites are constant. This implies that the vector of reaction rates, r∈ℝn, satisfies
(1)Nr=0,
i.e., it lies in the nullspace of N. In FBA and many other applications, not only is the steady-state condition taken into account, but also reaction reversibilities and general (lower and upper) bounds on reaction rates of the type
(2)$$lb_{i}\leq r_{i}\leq ub_{i}.$$

Setting a reaction with index i irreversible ($lb_{i}=0$) reflects assumptions about the thermodynamic properties of this reaction under physiological conditions. Other common flux bounds are substrate uptake limits or a lower bound on an ATP-hydrolyzing reaction that reflects ATP requirements for non-growth-associated maintenance. Beyond (1) and (2), general inequality constraints of the form
(3)Ar≤b,
with appropriate matrix A and vector b**,** can be easily embedded in an FBA problem, for example, to incorporate an upper limit for the total amount of enzymes that are needed to catalyze the fluxes contained in the rate vector [12,13]. The relations (1)–(3) give rise to a system of linear inequalities and its solution space forms a (flux) polyhedron. FBA problems contain, in addition, a linear objective function, such as the maximization of the growth rate, which is represented by a vector c and optimized subject to constraints (1)–(3):(4)$$max\;c^{T}r\;s.t.\;\left(1\right)-\left(3\right).\;$$

FBA problems constitute linear programs (LPs) for which a variety of solvers are available. We assume that the base LP (4) is feasible, i.e., there are no conflicting constraints in Equations (1)–(3) and at least one rate vector r exists that satisfies all these constraints. For metabolic models without fixed (known) rates, this is normally the case.

In the context of MFA, simple constraints are added to integrate values of known fluxes. The latter typically include measurements of exchange rates of substrates and products, but may also reflect environmental conditions or biological knowledge. For example, the pyruvate dehydrogenase (PDH) in *E. coli* is inactive under anaerobic conditions and the PDH flux as well as oxygen uptake can thus be set to zero when simulating anaerobic conditions. With F⊂{1…n}, we denote the indices of all reactions that have a fixed (known) flux and introduce equalities clamping these reaction rates to their known values:(5)$$r_{i}=f_{i},\;\;\;\;\;\;\forall i∊F.$$

We denote the number of known reaction rates with k=|F|. In principle, (5) could also be expressed via (2), by setting $lb_{i}=ub_{i}=f_{i}$. To keep things simple, we assume here that $lb_{i}\neq ub_{i}$ for all reactions i, i.e., the bounds of all reaction rates define a range of values and the truly fixed reaction rates are separately expressed via (5). Adding constraints (5) to the metabolic system (1)–(3) may now render the FBA problem (4) infeasible and, in this work, we discuss how these inconsistencies can be detected and resolved to proceed afterwards with a feasible system. 

We first review the techniques used to analyze and resolve infeasibility in classical MFA, which is solely based on constraints (1) and (5). This will later help us to relate the terminology and methods of classical MFA to the more general case of balancing infeasible FBA scenarios.

### 2.2. Classical MFA

In this section, we initially assume that the metabolic network has no conservation relations, i.e., the stoichiometric matrix has full rank (rank(N)=m). A necessary minor modification for the general case with conservation relations will be explained in Section 2.3. 

Classical MFA only considers constraints (1) and (5) and integrates them by splitting up the stoichiometric matrix and the vector of reaction rates into known and unknown parts:(6)$$N_{U}r_{U}+N_{F}r_{F}=0$$
(7)$$N_{U}r_{U}=-N_{F}r_{F}.$$

U={1,…,n}\F is the set of reactions with an unknown rate, rF and rU comprise the known and unknown reaction rates, respectively, and NF and NU are the corresponding stoichiometric submatrices for rF and rU, respectively. Since rF contains the known (fixed) values, (7) can also be rewritten as:(8)NUrU=z
with z=−NFrF. Equation (8) represents a classical system of linear equations with x=|U|=n−k unknown variables and *m* equations (n: number of reactions; m: number of metabolites; k: number of known reaction rates). Importantly, as discussed in the context of MFA in [9,11], every system (8) can be characterized via its determinacy and its redundancy, two key properties that both depend on the rank of NU. Regarding determinacy, a system is either determined (rank(NU)=x), meaning that all reaction rates are uniquely determined, or underdetermined (rank(NU)<x), implying that not all reaction rates are uniquely calculable because there are linear dependencies in the columns of NU. Accordingly, x−rank(NU) quantifies the degrees of freedom of a flux scenario (which equals the dimension of the nullspace of NU). Genome-scale networks are usually highly underdetermined, i.e., they have many more reactions than metabolites (m≪n) and, even with measurements of some exchange fluxes (or some assumptions), one still has rank(NU)≤m≪x. In metabolic core models, measurements (possibly in combination with knowledge about inactive fluxes) may bring the number of unknowns x closer to m (and thereby closer to the rank of NU). Generally, even if a genome-scale or core system is underdetermined, some (but not all) reaction rates may still be uniquely calculable in a given flux scenario. Reaction rates, whose value can be uniquely determined (with methods shown below), can be identified by analyzing an (arbitrary) basis of the nullspace of NU. Let KU denote the (kernel) matrix containing the vectors of a nullspace basis of NU. The rate of a reaction i is uniquely calculable if the i-th row of KU exclusively contains zeros, which implies that the i-th entry of all vectors of the nullspace of NU is zero [11].

The second key property of system (8) is redundancy. A system is redundant if it contains linear dependencies between some rows (metabolites) of $N_{U}$, i.e., if $rank\left(N_{U}\right)<m$, otherwise $rank\left(N_{U}\right)=m$ and it is non-redundant. The number of degrees of redundancy, degR, can be calculated as
(9)$$degR=m-rank\left(N_{U}\right).$$

As a sub-categorization, if the system is redundant, it is either consistent (feasible) or inconsistent (infeasible), whereas non-redundant systems are always consistent. Consistency or feasibility means that there is at least one vector $r_{U}$ solving Equation (8). In terms of linear algebra, system (8) is consistent if vector z in Equation (8) is in the image (column space) of matrix $N_{U}$. Before clarifying how we can detect the (known) rates that cause the redundancies in a redundant system, we consider a general solution for the MFA scenario (8), which is given by
(10)$$r_{U}=N_{U}^{\#}z.$$

$N_{U}^{\#}$ is the x×m Moore–Penrose pseudoinverse of $N_{U}$ existing for every possible matrix $N_{U}$ [14]. Importantly, (10) is the classical least-squares solution for (8) known from linear regression, i.e., it minimizes the squared error
(11)$${\left(N_{U}r_{U}-z\right)}^{T}(N_{U}r_{U}-z).$$

Using (10) and $z=-N_{F}\cdot r_{F}$ and by defining the redundancy matrix R as
(12)$$R=N_{F}-N_{U}N_{U}^{\#}N_{F}$$

We can rewrite (11) as
(13)$${\left(Rr_{F}\right)}^{T}\left(Rr_{F}\right)=r_{F}{}^{T}R^{T}Rr_{F}.$$

If the MFA scenario is feasible, then (11) and (13) yield zero or, equivalently,
(14)$$Rr_{F}=0.$$

In a non-redundant system, R=0 always holds implying consistency for any instance of $r_{F}$. In a redundant system, one has R≠0 and it then depends on the values of the measured rates in $r_{F}$ whether or not the system is feasible. The system is infeasible if $Rr_{F}\neq 0$ and the solution $r_{U}$ obtained with the pseudoinverse (10) then ensures minimal distance of $N_{U}r_{U}$ from vector z (distance measured as Euclidean norm $|\left|\;\right|{|}_{2}$ as expressed by Equation (11)). Note that the least-squares solution for (8) might not be unique, if the system is underdetermined ($rank\left(N_{U}\right)<x$). Solution (10), based on the Moore–Penrose pseudoinverse, is then still unique in the sense that the delivered vector $r_{U}$ has a minimal length (again in Euclidean norm, i.e., it minimizes $r_{U}{}^{T}r_{U}$) of all least-squares solutions. In underdetermined systems, as already mentioned above, the uniquely determined rates from $r_{U}$ correspond to zero rows in the kernel matrix (containing a basis of the nullspace of $N_{U}$) [11].

Since the absolute value of $Rr_{F}$ quantifies the error of solution (10) with respect to Equation (8), one additional important piece of information can be derived from the redundancy matrix R: all reactions in $r_{F}$ that have any non-zero value in the corresponding column in R are involved in the posed redundancies and are thus responsible for potential inconsistencies (we call these known reaction rates the redundant rates).

We illustrate the introduced terms with the example network shown in Figure 1. Since determinacy and redundancy are independent (binary) properties, 2 × 2 = 4 possible cases can arise and at least one example flux scenario is shown in Figure 1 for each of these four cases. The network has n=10 reactions and m=6 metabolites giving rise to a 6×10 stoichiometric matrix N, which has a full rank of 6 (no conservation relations). From linear algebra, it is clear that we need at least 10−6=4 measured reaction rates to obtain a determined system. Hence, scenario S1, in which the rates of reactions R1 and R4 are known, must be underdetermined and not all unknown rates can be determined. However, it is easy to observe that the reactions R7 and R10 must have the same rate as reaction R4 to fulfill the steady-state condition. Hence, despite being an underdetermined system, some rates are uniquely determined. 

The second scenario, S2, is typical for realistic MFA applications: all (four) exchange rates were measured. In well-posed systems, this results in a redundant system, since knowing all exchange fluxes but one always allows, due to mass balances, the calculation of the unknown exchange flux (in the example network, it must hold that $r_{R1}+r_{R2}=r_{R3}+r_{R4}$). Accordingly, this scenario is redundant and an analysis of the redundancy matrix reveals that all known reaction rates contribute to this redundancy, and that the system is infeasible. The scenario is also underdetermined and the analysis of the nullspace of $N_{U}$ shows that the rates of reactions R7, R9, and R10 are uniquely determined and can be taken from the calculated least-squares solution (10). It can be observed that the solution, due to infeasibility, does not fulfill the steady-state condition. For example, with the calculated fluxes for R7 and R10, the metabolite D is not in steady state. However, the pseudoinverse solution ensures minimal deviation from the steady state. Scenario S3 is identical to S2, except that now a value of 4 is given for the rate of R3. This scenario is again redundant, but this time feasible, as also indicated by the fluxes around the metabolite D which is now in steady state. However, such a case practically never occurs in reality, since measurements are overlaid with noise and it is extremely unlikely that a redundant system (with real measurements) exactly fulfills the consistency condition (14). 

Scenario S4 extends scenario S2 with the assumption that reaction R6 is inactive (rate is zero). The system is still redundant, but now all rates are uniquely determined (the system is determined). A system that is redundant and determined is often called overdetermined. Scenario S4 is still infeasible, but the redundancies (and inconsistencies) are again only posed by reactions R1–R4, whereas reaction R6 adds no redundancy to the system. The steady-state condition is violated for all internal metabolites but, still, it is the best solution that can be found for Equations (7) and (8) as it minimizes the (squared) error in (11). Finally, scenario S5 is identical to S4, except that we assume that we had not measured the rate of reaction R2. This system is not redundant, but determined and all rates (including for R2) can be uniquely calculated. Only in such a non-redundant and determined system does the Moore–Penrose pseudoinverse become a square matrix coinciding with the (then existing) inverse of $N_{U}$. The system is feasible, i.e., all steady-state conditions are satisfied in the found solution. This example shows that leaving out some measurements can make an infeasible (redundant) system feasible, but we would lose important information about the possible inconsistencies in our measurements, which, if larger in size, could point to systematic measurement errors or missing reactions in our network model.

### 2.3. Conservation Relations

In the considerations above, we assumed that there are no conservation relations (CRs) in the metabolic network, which refer to linearly dependent rows (metabolites) in the stoichiometric matrix N. Hence, a reaction network contains CRs, if the stoichiometric matrix does not have full rank, rank (N)<m, and we denote the number c of linearly independent CRs by c=m−rank (N) (equivalently, c is the dimension of the left nullspace of N). CRs impose redundancies already in the initial system (1) but, since (1) is a homogeneous system with a zero vector on the right-hand side, these redundancies are always consistent. For MFA, one could remove dependent rows and thus the CRs from the stoichiometric matrix before performing further calculations. If they are kept, we only need to take into account that the rank deficiency will also be retained in any given MFA scenario, but that these dependent rows will not induce inconsistencies. Therefore, for the classification of redundancy of any MFA scenario (with or without CRs), we now state that the system is redundant if $rank\left(N_{U}\right)<m-c$, which is equivalent to $rank\left(N_{U}\right)<rank\left(N\right)$, hence, only if new row dependencies occur in $N_{U}$ due to the given set of measurements. Likewise, the number of degrees of redundancy for the general case is calculated as:(15)$$degR=m-rank\left(N_{U}\right)-c=rank\left(N\right)-rank\left(N_{U}\right).$$

Everything else remains unaffected, in particular, the redundancy matrix and the redundant rates can be determined as before.

### 2.4. Correcting Measured Rates to Make Inconsistent Systems Feasible

Scenarios S2 and S4 in Figure 1 illustrated that the least-squares solution (9) based on the pseudoinverse delivers in inconsistent systems solutions for rU that inevitably violate the steady-state conditions for some metabolites. Here, it would be desirable to first correct the measured rates to obtain a feasible system and to only then determine the (uniquely calculable) rates that obey the steady-state conditions. There are many ways to correct inconsistent measurements to obtain a feasible scenario, but it is reasonable to demand that the introduced changes are minimal with respect to a certain measure. Here, it would be desirable if we could include information on the noise of the respective measurements. A suitable approach, based on a weighted least square optimization, can be constructed as follows [7]. We aim to correct the values fi of the measured/known reaction rates ri by introducing a correction variable δi∊ℝ
(16)ri=fi−δi ,   ∀i∊F,
so that the system (7) becomes feasible. From all possible solutions for δi, we look for the one that minimizes the weighted squares of all corrections:(17)min∑i∊Fwiδi2s.t. (1) and (16).

For the weights $w_{i}$, we choose the reciprocal values of the measurement variance $\sigma_{i}^{2}$ of the known rates: $w_{i}=1/\sigma_{i}^{2}$. The solution for this minimization problem is given by
(18)$$\delta =WR_{r}^{T}{\left(R_{r}WR_{r}^{T}\right)}^{-1}R_{r}r_{F}\;,$$
where W is a diagonal matrix containing the reciprocal values of the $w_{i}$ (i.e., the $\sigma_{i}^{2}$ of the measurements) on its diagonal, whereas $R_{r}$ is the reduced redundancy matrix, which can be obtained from the redundancy matrix R in Equation (12) by removing dependent rows from the latter. If the measurement variances are not known, one may either choose an identical weight $w_{i}=1$ for all reactions (resulting in an identity matrix for W), or set $w_{i}=1/\left|f_{i}\right|$ assuming that the measurement noise is correlated with the magnitude of the measured value. With the calculated corrections $\delta_{i}$, the reaction rates can be corrected via (16) yielding a feasible system from which the unknown rates (as far as uniquely calculable) can then be obtained as before from the pseudoinverse solution in Equation (10).

Although both Equation (18) as well as the solution based on the pseudoinverse in Equation (10) are least-squares solutions, they are fundamentally different: The former minimizes the deviation from a steady state under the consideration of the given reaction rates, whereas the second minimizes the (weighted) changes in the measured rates necessary to obtain a feasible system fulfilling the steady-state condition. This is illustrated by scenario S6 in Figure 1, which takes as input the same measured values as in S4, but this time it is solved via Equation (18) using $w_{i}=1$ for all known rates. The four redundant rates of R1–R4 (but not the non-redundant rate of R6) are adjusted to obtain a feasible system and, from the corrected scenario, the solution for the unknown rates is computed now obeying the steady-state condition for all internal metabolites.

## 3. Results

### 3.1. Infeasibility in FBA Scenarios with Known Fluxes

Classical MFA cannot consider any other constraints than the steady-state condition (1) and the measured reaction rates (5). Even when considering only simple reversibility constraints (by using $lb_{i}\geq 0$ for all irreversible reactions in (2)), classical MFA may deliver wrong results and infeasibility may even remain undetected. For example, in the underdetermined scenario S7 in Figure 1, the algebraic solution via (10) would indicate that the scenario is non-redundant ($rank\left(N_{U}\right)=m$) and thus feasible, and that the rates of reactions R7 and R10 are 6 (which follows since the rate of R4 is 6). However, since all reactions except R2 and R8 are irreversible in the example network in Figure 1, no feasible solution can exist for this scenario, because the rate of R4 can then not be larger than the flux through R1. Accordingly, applying FBA (with any objective function) would result in an infeasibility message of the solver.

We therefore consider in the following an FBA scenario consisting of the FBA base constraints (1)–(3), the flux measurements (5), and the given FBA objective function from (4): (19)$$max\;c^{T}r\;s.t.\;\left(1\right)-\left(3\right),\left(5\right).\;$$

As previously stated, we assume that the base constraints (1)–(3) form a feasible system, a property that is fulfilled in well-posed network models (the general case, where inconsistencies in the base constraints could also be detected/corrected, will be discussed in Section 3.4). Assume now that fixing the known/measured fluxes (5) and adding these constraints to the base FBA constraints (1)–(3) leads to an infeasible system, which will be reported by the LP solver when trying to solve (19). The goal is now to identify and resolve the inconsistencies to obtain a feasible system. Because the FBA’s objective function bears no relevance on the feasibility of the LP (19), we remove it for the following considerations. This would also reflect the situation where we are interested to (only) identify the uniquely determined rates that follow from the extended MFA scenario with constraints (1)–(3) and (5), without having an objective function (as in scenario S7 in Figure 1). 

### 3.2. Weighted Least-Squares Solution via Quadratic Optimization

We search for minimal corrections of the flux measurements (5) that make the system feasible. We know that such a correction exists, as we assume that the base system (1)–(3) is feasible. We can follow a similar strategy as we did for the (algebraic) least-squares approach (Equations (16) and (17)) of classical MFA. We again add a correction term (or slack variable) δi∊ℝ to each of the fixed reaction rates
(20)ri=fi−δi ,  ∀i∊F 
and formulate an optimization problem that minimizes the (weighted) sum of the squared corrections
(21)$$min\underset{i∊F}{\sum }w_{i}\delta_{i}^{2}\;s.t.\;\left(1\right)-\left(3\right)\;\mathrm{and}\;\left(20\right).$$

This approach mimics the weighted least-squares approach of classical MFA (17), but the difference here is that this optimization is subject to the full set of constraints (1)–(3) and (20), for which no analytical solution as in Equation (18) exists in general. In this setting, (21) becomes a quadratic program (QP), which is an LP extension that allows for quadratic terms in the objective function, while the constraint must still be linear. In general, the objective function (over variables x∈ℝn) of a convex QP has the form
(22)minimize  xTQx+cTx+d,
where Q∈ℝn×n is a symmetric positive semidefinite matrix, c∈ℝn, and d is a constant [15]. In our case, x is a vector containing the reaction rates as well as the correction terms δi. Furthermore, we have c=0, d=0, and Q is a matrix containing zeros, except some diagonal elements containing the weights wi associated with the δi. Since all weights can here be assumed to be positive (see below), Q is positive semidefinite as demanded. The computed corrections δi for the known ri in (20) are unique at the minimum of the objective function (21). However, the values of the undefined reaction rates in r are, in general, not unique if the system is underdetermined.

Regarding the weights $w_{i}$ in (21), we can use the same three weighting schemes as proposed for the weighted least-squares approach in classical MFA:

W1: If the variances of measurements ($\sigma_{i}^{2}$)are available, these could be used for the weights: $w_{i}=1/\sigma_{i}^{2}$.

W2: If the measurements variances are unavailable, one could set $w_{i}=1/\left|f_{i}\right|$ so that a deviation from a flux of larger magnitude weighs less than the same deviation from a flux of lower magnitude. This assumes that the measurement noise is correlated with the magnitude of the measured value.

W3: As the simplest approach, one could choose equal weights for all corrections: $w_{i}=1$.

For W1 and W2, one should use large positive weights if a rate was measured (or assumed) to be zero to set high penalties for corrections in those reaction rates. In case one is certain that a rate is zero, the corresponding reaction can also be removed from the network prior to the calculation or a weight of infinity could be used.

We note that the corrections $\delta_{i}$ obtained with the QP (21) are identical to the least-squares solution of classical MFA (18), as long as the constraints (2) and (3) do not matter for the infeasibility (or if they would be dropped). For example, in scenario S8 in Figure 1, we reconsider scenario S6, but this time apply the delineated QP procedure. This yields exactly the same corrections for the known rates, even if the reversibility constraints are included in the QP. Classical MFA uses, after computing and applying the minimal corrections, the pseudoinverse solution (10) together with a nullspace analysis of $N_{U}$ to identify the values of the uniquely determined (unknown) rates. In contrast, in FBA-based MFA a flux variability analysis (FVA, [16]) can be performed to the corrected system to determine the minimum and maximum flux for each reaction. In particular, this reveals the fluxes that are uniquely determined from the (corrected) predefined rates and, in addition, deliver the admissible flux ranges for the remaining fluxes. Scenario S9, which reconsiders scenario S7 with the QP approach (21) followed by FVA, now handles the inconsistencies between the given values and the reaction reversibilities properly, and subsequently delivers the values of uniquely determined rates (R5, R6, R7, R8, R10) as well as the valid ranges of the still unknown rates. Note that even if we would start with the corrected system (R1 = R4 = 5) in scenario S9, taking the pseudoinverse in combination with the nullspace analysis of $N_{U}$ would deliver the result that only the rates of R7 and R10 are uniquely determined. The fact that R5, R6, and R8 are also determined in this particular case, follows from the irreversibility of R5 and R6 and can therefore only be deduced from FVA, not from the classical MFA approach.

### 3.3. Using Linear Optimization to Correct Flux Measurements in Infeasible FBA Systems

The QP-based approach for resolving inconsistencies goes beyond classical LP problems considered in FBA and requires a suitable QP solver. Furthermore, although the complexity of solving convex QP problems is polynomial [15], the computation time for a QP is typically somewhat higher than for an LP, and the applications of a QP in genome-scale models may become more elaborate (see Section 3.7). An alternative, solely LP-based approach can be constructed as follows. Instead of a single correction term δi∊ℝ used in (20), we now introduce two non-negative slack variables δi+≥0, δi−≥0 for each measured/known rate
(23)ri=fi+δi+−δi−,  ∀i∊F.

As the objective function, we minimize the weighted sum of the slacks needed to make the system feasible
(24)$$min\underset{i∊F}{\sum }w_{i}\left(\delta_{i}^{+}+\delta_{i}^{-}\right)\;s.t.\;\mathrm{to}\;\left(1\right)-\left(3\right)\;\mathrm{and}\;\left(23\right).$$

Effectively, this optimization problem minimizes the weighted sum of absolute changes and the use of two non-negative slack variables in (23) and (24) allows for the linear representation. Furthermore, note that in any optimal solution obtained with (24), at least one of the two correction terms $\delta_{i}^{+}$ and $\delta_{i}^{-}$ is zero. Regarding the weights $w_{i}$, we have the same possible choices as for the QP-based approach. 

It should be noted that, in contrast to the QP method, the LP-based correction approach (24) may have non-unique solutions for the slack variables. Consider a simple inconsistent scenario in the network in Figure 1 with the predefined rates R4 = 2 and R10 = 4. If we use weighting scheme W3 (equal weights $w_{i}=1$) in (24), all corrections that lead to R4 ∈ [2,4] and R4 = R10 are valid optimal solutions with a minimal objective value (sum of the slack variables) of two. Hence, there are infinitely many solutions and the found values of the slacks depend on the solver and its solution method. The only QP-based solution in this case would result in R4 = R10 = 3. However, if measurement variances ($w_{i}=1/\sigma_{i}^{2}$; weighting scheme W1) or absolute flux values ($w_{i}=1/\left|f_{i}\right|$; weighting scheme W2) are used for the weights, the results are much more likely to be unique, because then the weights often have distinct values. For example, if we used $w_{i}=1/\left|f_{i}\right|$ in the example above, the unique solution for (24) would be R4 = R10 = 2 (whereas the unique QP solution is then R4 = R10 = 2.67). The choice of the weights, but also the used correction method, thus influence the outcome of the correction and thereby also impact the results of the subsequent analyses with the corrected scenario. There is no general rule for which method is to be preferred. In the related field of linear regression, if measurement variances are available (and used as weights), weighted least-squares approaches are often used as they lead to a unique solution and provide some theoretical results about statistical properties if the measurement errors are normally distributed [17]. However, one should also be aware of the fact that the quadratic terms in the QP (21) penalize large deviations more strongly. Although this ensures a unique solution for the corrections, it also implies that typically all rates involved in the inconsistency will receive a correction. In realistic applications, we found that the LP approach with $w_{i}=1/\left|f_{i}\right|$ also delivered reasonable results and may thus provide an alternative to the QP approach, especially if no QP solver is available or if the measurement variances are not known.

### 3.4. Allowing Minimal Corrections of Other Constraints to Make Infeasible FBA Systems Feasible

Thus far, we assumed that the fixed (measured/known) fluxes in (5) caused the inconsistencies and searched for suitable (minimal) corrections of these fluxes to make the system feasible. However, one may also consider the other constraints (1)–(3) as possible sources of the inconsistencies, and thus as targets for corrections. The procedure is analogous as for correcting the fixed fluxes. One first introduces slack variables on the right hand-side of the respective constraints. For each of the m rows in the equality-based steady-state constraint (1), we would again either use a single slack δi∊ℝ (in combination with the least-squares/QP approach (21)) or two non-negative slacks (for the LP-based approach (24)), whereas introduction of a single non-negative slack as a correction term suffices for each inequality constraint in (2) and (3). For example, the flux bound constraints (2) would be rewritten as
(25)ri≤ubi+δiubri≥lbi−δilb
with δiub≥0, δilb≥0. Subsequently, weights can be assigned to each introduced slack variable and the QP (21) or the LP-based (24) correction procedure is applied to find the optimal solution. Interestingly, if we consider the system with constraints (1)–(3) and (5), and allow corrections (with equal weights) only for the steady-state condition (1), the QP-based solution for the corrections would be identical to the pseudoinverse solution (10), as long as the inconsistencies arise solely due to algebraic redundancies in the measurements (and not in combination with flux bounds (2) or other constraints in (3)). Hence, using this procedure with a QP-based optimization for scenario S4 in Figure 1 delivers the same results as shown for the pseudoinverse. 

Normally, the (exclusive) correction of fixed/measured fluxes is most relevant for realistic MFA/FBA applications. Nevertheless, there might be cases in which the correction of other constraints is of interest, for example, (a) to identify reactions that are constrained to be irreversible but are in reality reversible, or (b) to resolve inconsistencies related to complex enzyme capacity constraints [12,13] that can be expressed via (3).

### 3.5. A Practical Guide for How to Proceed with Infeasible FBA Problems

As a summary, in the following, we provide a step-by-step procedure for how to analyze and resolve infeasibility in FBA scenarios:

(1)Detect (in)feasibility: infeasibility of a metabolic flux (balance) scenario can or will be detected by an associated error message of the LP solver when performing an FBA optimization or an FVA. If this infeasibility is not a consequence of fixing some reaction rates, i.e., if the base system (1)–(3) is already infeasible, use the methods explained in the previous section for resolving inconsistencies in these constraints.(2)If not yet performed, identify all reactions with a fixed rate (e.g., by searching for reactions in the model where lower and upper bound are identical).(3)It is recommended to check via Equation (15) whether there are algebraic redundancies in the defined scenario (solely related to the steady-state constraint) and, if so, to find out which of the fixed reaction rates induce this redundancy (via the redundancy matrix (12)). This can be very helpful to find possible sources of mutually inconsistent rates (but it does not provide a (complete) explanation if constraints (2) and (3) are also involved in the inconsistency).(4)Decide on which optimization approach (QP (21) or LP (24)) and which of the three weighting schemes, W1–W3, are to be applied. Recommendation: if the measurements variances $\sigma_{i}^{2}$
are known and if a suitable QP solver is available, then use a QP with $w_{i}=1/\sigma_{i}^{2}$; otherwise, choose the LP approach with $w_{i}=1/\left|f_{i}\right|$. In the latter case, the $w_{i}$ may be manually (re)adjusted to increase knowledge about the (un)certainty of the fixed rates. Generally, large weights ($w_{i}$ > 1000) should be used for rates fixed at zero ($f_{i}=0)$. 

(5)Compute the corrections with the respective optimization approach and analyze them to identify the given rates that were assigned the largest changes and have thus caused the largest inconsistencies.(6)Apply the corrections and compute with the balanced (now feasible) system the solutions for the original FBA/FVA problem. In particular, FVA can be used to identify unknown rates that are uniquely determined from the measured rates (FVA delivers identical lower and upper bounds for those rates). Generally, when performing FBA or FVA in the corrected system, one may face numerical problems: the precision of the calculated (and applied) corrections might not be sufficient enough for the solver used in the subsequent FBA/FVA optimizations, again resulting in an infeasibility message of the solver. If this happens to be the case, it is advised to use a slightly lower precision for the subsequent optimizations. In the example applications described in Section 3.7, we did not encounter such a problem.

### 3.6. Implementation in CellNetAnalyzer and CNApy 

Previous versions of the MATLAB toolbox CellNetAnalyzer (CNA) [18,19] already provided methods for balancing infeasible MFA/FBA scenarios, which have been largely revised in the latest version (2022.1) to embed the new methods presented herein. In particular, the LP-based (supported solvers: CPLEX, glpk, and MATLAB’s linprog) as well as the QP-based (supported solvers: MATLAB’s quadprog and CPLEX’s cplexqp) approaches for balancing infeasible FBA systems have been implemented, and can both be used with any of the three proposed weighting schemes. Classical MFA calculations based on the pseudoinverse (Equations (10) and (18), respectively) can still be performed and used for comparison. CNA can be downloaded via http://www2.mpi-magdeburg.mpg.de/projects/cna/cna.html (accessed on 22 June 2020). 

Likewise, these methods have been implemented in the latest release of CNApy, a recently published Python toolbox [20]. CNApy has a similar scope as CNA, but provides significantly enhanced GUI features. Supported solvers are CPLEX, glpk, and gurobi for the LP-based approach, and CPLEX and gurobi for the QP approach. CNApy can be obtained from: https://github.com/cnapy-org/CNApy (accessed on 22 June 2020).

### 3.7. Relevant Examples from Core and Genome-Scale Models of Escherichia coli

The methods introduced herein are especially useful to treat infeasibility in FBA systems arising from measured or known fluxes. How likely is it in practice that a network model becomes infeasible if flux measurements are added to the network constraints? Clearly, this depends on (i) network size and structure, (ii) the set of measured fluxes, and (iii) the type of included constraints in (3). Fluxes that can often be determined for microorganisms in dedicated experiments are the exchange rates of (carbon) substrates, oxygen, carbon dioxide, hydrogen and of carbon (by)products, as well as the growth rate. Fixing these fluxes in the (compressed) *Escherichia coli* core model 2 (ECC2comp), a model of the central metabolism of *E. coli* comprising 82 reactions [21], already results in two (algebraic) degrees of redundancy (degR=2 in Equation (15)). These redundancies arise from carbon and redox balances in combination with the steady-state condition (1). Hence, with imprecise measurements, one obtains an infeasible scenario and further inconsistencies may arise due to reversibility or other constraints. Analyzing the magnitudes of the corrections required to make the system feasible provides important information about the consistency of the model and the observed fluxes. 

Then, one may apply FVA to investigate whether uniquely determined rates result from the given scenario. However, even in core models and even if further constraints are added (such as non-growth associated maintenance (NGAM) demand of ATP, enzyme capacity constraints, etc.), typically only few of the unknown internal rates become uniquely determined. Here, FBA can be used to focus on optimal flux distributions. If the growth rate has been measured and added as a constraint, the classical FBA objective function of growth maximization would not be meaningful anymore. A reasonable assumption is then to assume that, under the given constraints, the cell has optimized the remaining fluxes towards ATP production, i.e., the NGAM reaction (usually denoted by ATPM) is maximized. This approach was also used in [22] after resolving the inconsistencies in the measured fluxes with an LP-based approach. 

Although less likely, infeasibilities can also arise in genome-scale models and, thus, methods to analyze and treat them become relevant in these large models as well. In particular, redundancies will appear in any well-posed stoichiometric metabolic model if all exchange fluxes relevant for the balance of a certain element (carbon, nitrogen, phosphate, sulfur, hydrogen, etc.) or for charge balance have been measured or fixed. For example, fixing the fluxes of all exchange reactions that transport carbon metabolites in the genome-scale *E. coli* model iJO1366 [23] indeed leads to a redundancy, although more than 800 degrees of freedom remain in this scenario. There might be realistic applications where some of the respective carbon exchange fluxes are known from measurements (growth rate, substrate uptake, standard (fermentation) products, CO_2_), while the others can be assumed to be zero. Again, the resulting redundancy and the magnitudes of the computed (minimal) corrections needed to make the system feasible can then deliver important insights on the overall carbon balance and point to possible modeling (or measurement) errors. 

Redundancies related to carbon balance are the most valuable, as the latter integrates a large number of fluxes in the metabolism; however, redundancies from other elemental or charge balances can be relevant and useful as well. In contrast to simple algebraic models of elemental balances [7,9], FBA models (with truly mass-balanced reaction stoichiometries) contain elemental balances implicitly and together with flux bounds (Equation (2)) and possibly further constraints (Equation (3)) they are less flexible, through which infeasibilities are more likely to appear. In fact, inconsistencies may also arise without any algebraic redundancy related to the steady-state constraint (1). For example, using a genome-scale model of *E. coli* with enzyme constraints [13] and providing only three measured rates for substrate (glucose) uptake, excretion of acetate (reflecting overflow metabolism), and biomass synthesis (growth rate) can easily result in infeasibilities, which are difficult to resolve intuitively or manually because all three depend on each other via the global enzyme capacity constraint.

We also tested the LP- and QP-based methods implemented in CellNetAnalyzer to resolve typical examples of infeasible scenarios in the compressed core (ECC2comp) and in the genome-scale (iJO1366) model of *E. coli.* In both networks, we considered three different infeasibility problems (P1–P3) with increasing complexity: P1—two strictly coupled reactions have inconsistent fluxes; P2—growth rate and all carbon exchange fluxes fixed (with inconsistent values); P3—one reaction rate within a given complete flux distribution (FBA solution for maximization of growth) is changed to an inconsistent value. For each problem in each network, we considered two weighting schemes (W2 and W3) and calculated the LP-based as well as the QP-based solution. We first used CPLEX’s (cplexlp for the LP-based and cplexqp for the QP-based approach) and then MATLAB’s solver (linprog for LP and quadprog for QP) with standard tolerances and parameters. The calculation via the LP-based approach could be successfully finished within one second (standard PC) for all tested problems in the core, as well as in the genome-scale model with both tested LP solvers. Regarding the QP-based solutions, all problems in both networks could again be solved within one second when using CPLEX’s QP solver with the “simplex” method. In contrast, while MATLAB’s QP solver succeeded the calculation of all example scenarios in the core network (again, in less than one second), only for one (out of six) tested genome-scale problems a QP-based solution could be found. In the other cases, the solver stopped with the message that it had reached the maximum number of iterations or that it encountered numerical problems (using other tolerances was also not successful). These results indicate that the LP-based solution may in some cases be more practical in large-scale models. 

## 4. Discussion

In this work, we presented and discussed methods useful to analyze and resolve infeasible FBA scenarios. We focused on the most relevant situation in which inconsistencies arise due to predefined fluxes, but also outlined the more general case where inconsistencies are caused by other linear constraints of an FBA scenario. We are not aware of any previous work that provides a systematic approach to treat and resolve inconsistent FBA scenarios with fixed fluxes and how this links to classical MFA. Abbate et al. [24] used adaptive flux variability analysis to find suitable interval representations of measured fluxes to turn an infeasible FBA scenario into a feasible one. However, such an interval representation is actually not required for resolving inconsistency and the method considers neither different weights nor quadratic terms for the corrections in the objective function. In the context of FBA, a popular method that uses quadratic programming is MOMA (minimization of metabolic adjustment) [25]. Despite some similarities, the focus is different: MOMA starts with an (entire) reference flux distribution obtained from an FBA optimization. Within this reference solution, one (or several) fluxes are set to zero corresponding to knockouts. QP optimization is then used to find minimal (quadratic) changes in the other fluxes to obtain a feasible solution, with the particularity that zero fluxes associated with knockouts must not be changed. In the (MFA-based) FBA scenario considered herein, we are usually given a few (measured) fluxes, which, if inconsistent, will be corrected to obtain a feasible solution. Additionally, different to MOMA, the changes can be weighted or a linear objective function can be used. In principle, MOMA can be seen (and run) as a special case of the QP-based correction approach where the fixed zero fluxes corresponding to knockouts obtain a very large (infinite) weight in the quadratic objective function.

Another small methodological overlap exists with FBA methods that seek to identify stoichiometric inconsistencies (such as mass-imbalanced reaction stoichiometries or blocked reactions) in metabolic models [26]. However, these inconsistencies are related to the particular structure and properties of the stoichiometric matrix (not to flux measurements or other flux constraints), and are relevant in the context of network reconstruction to obtain physically and chemically valid models. In fact, a model with (undetected) mass-imbalanced reaction stoichiometries may be feasible in the context of a specific FBA scenario. 

We presented and discussed two major approaches for treating infeasibility: the QP-based approach minimizing the (weighted) squares of changes, and the LP approach minimizing the sum of (weighted) changes. In the field of linear programming, a variety of further methods has been developed to deal with infeasibility in systems of linear equalities and inequalities [27]. For example, as another option not discussed so far, one could minimize the number of non-zero changes needed to make the system feasible (instead of the sum of (squared) changes). This could be achieved with a mixed-integer linear program (MILP). Such a scheme might be useful during network exploration with many rates fixed to reveal the smallest number of modifications necessary to make the FBA feasible again. However, the problem of finding a minimal number of required changes is NP-hard [27], although state-of-the-art MILP solvers may be able to find optimal MILP solutions in genome-scale networks. Generally, such cardinality-based optimizations are also relevant for other constraint-based modeling techniques and a related algorithm is available at [28].

Another possible extension is to analyze in more detail the structural sources of the detected infeasibility, for example, by identifying smallest (irreducibly) inconsistent subsets of constraints (IIS [27]). To make an infeasible system feasible again, one constraint from each IIS needs to be removed. However, the results from an IIS analysis are rather abstract and may not directly point to the actual source of potential modeling or measurement errors.

## Figures and Tables

**Figure 1 metabolites-12-00585-f001:**
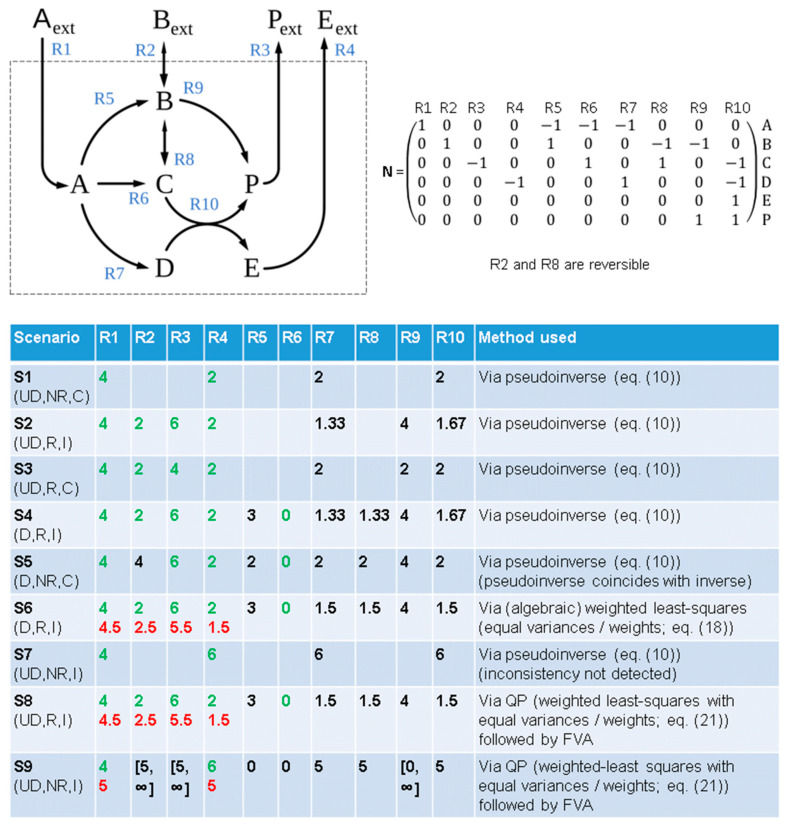
Example network with different MFA scenarios. Green values mark known (“measured”) reaction rates and red values indicate computed minimal corrections for the measured rates that make the scenario feasible. Black values correspond to reaction rates that can be uniquely determined from the given rates or, in case of scenario S9, they reflect identified ranges of feasible fluxes as determined by FVA. Each scenario is characterized by determinacy (D: determined; UD: underdetermined), redundancy (R: redundant; NR: non-redundant), and consistency/feasibility (C: consistent, I: inconsistent).

## Data Availability

Data is contained within the article.

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
