# Peer review of "Analyzing and Resolving Infeasibility in Flux Balance Analysis of Metabolic Networks"

_metabolites, 2022, doi:10.3390/metabo12070585_

Round 1
Reviewer 1 Report
The authors present an analysis which type of algebraic flux scenarios can occur when experimental data is combined with metabolic models. Their report is conceptually important because it provides a thorough discussion of the possible inconsistencies when bringing data to models and provides strategies how to resolve the inconsistencies. This is relevant both to computational scientist for insights into the network architecture but also for experimentalists because reactions are identified for which the measured fluxes deviates from a consistent flux set. The theoretical approach is appropriately described and a practical implementation is provided via online tools.
Major points:
1. The approach seems similar to the MOMA optimization (Segre et al., 2002, doi:10.1073/pnas.232349399) as well as Martin et al. (2015, doi:10.1371/journal.pcbi.1004363) and the authors should explain how their approach relates to the two published strategies.
2. Line 545ff: There is no data presented for the results section 'Relevant examples from....'. This section could rather be a discussion. The two paragraphs are too long and could be broken and simplified.
3. The results section could start at line 209 with the analysis of the scenarios and also Eq. 17 is as much a result as Eq. 21, this also automatically moves the only figure into the results section.
4. Line 276: It would be constructive to have an example at hand during the development of the redundancy scenario.
Minor points:
1. Some Equations would benefit from a repetition of variable names close by: 'A' in Eq. 3, 'cT' in Eq. 4, and 'n', 'k', 'm' in line 151.
2. Line 232: R3 uses a value of 4 (not 2) for S3 in the Figure 1.
3. Line 394: Which number should be chosen for an infinite weight? It is counter intuitive that inactive reactions are assigned to a high weight (experimental uncertainty).
4. Line 541 CNapy-> CNApy
Author Response
Reviewer #1:
The authors present an analysis which type of algebraic flux scenarios can occur when experimental data is combined with metabolic models. Their report is conceptually important because it provides a thorough discussion of the possible inconsistencies when bringing data to models and provides strategies how to resolve the inconsistencies. This is relevant both to computational scientist for insights into the network architecture but also for experimentalists because reactions are identified for which the measured fluxes deviates from a consistent flux set. The theoretical approach is appropriately described and a practical implementation is provided via online tools.
We thank the reviewer for this positive assessment.
Major points:
1. The approach seems similar to the MOMA optimization (Segre et al., 2002, doi:10.1073/pnas.232349399) as well as Martin et al. (2015, doi:10.1371/journal.pcbi.1004363) and the authors should explain how their approach relates to the two published strategies.
Our QP (but not the LP) approach has some similarities with MOMA, yet, the focus is different: MOMA starts with an (entire) reference flux distribution obtained from an FBA optimization. Within this reference solution, one (or several) fluxes are set to zero corresponding to real knockouts. QP optimization is then used to find minimal (quadratic) changes in the other fluxes to obtain a feasible solution, with the particularity that zero fluxes associated with knockouts must not be changed. In our (MFA-based) FBA scenario, we are usually given a few (measured) fluxes, which, if inconsistent, will be corrected to obtain a feasible solution. Also different, the changes can be weighted or a linear objective function can be used. In principle, MOMA can be seen as a special case of our QP-based approach where the zero fluxes corresponding to knockouts obtain a very large (infinite) weight. In fact, the described QP-based approach for resolving infeasibility can also be used to calculate MOMA solutions. We now mention the relationship to MOMA in the discussion section.
The second paper mentioned by the reviewer seeks to integrate data/fluxes from 13C flux analysis into genome-scale models to further constrain the latter. However, we could not find methods related to ours that seek to resolve infeasibilities in FBA-related studies. In that paper, errors are minimized with least-squares methods during 13C flux analysis, but (non-linear) 13C flux analysis cannot directly be compared with our FBA-based techniques.
2. Line 545ff: There is no data presented for the results section 'Relevant examples from....'. This section could rather be a discussion. The two paragraphs are too long and could be broken and simplified.
We agree that parts of the section 3.7 (“Relevant examples from …”) are more in the style of a discussion. However, we would nevertheless like to keep this section in the Results chapter because several discussed aspects are indeed specific to the respective models (e.g. two degrees of redundancy in the ECC2 model if all measurable exchange fluxes are provided). In addition, as suggested by one reviewer, we now also extended this section with further specific information on tests of the LP- and QP-based approaches in the core and in the genome-scale model. We therefore feel that integrating this extended section within the Discussion chapter would blow up the latter (which was also extended due to suggestions from another Reviewer). However, we followed the suggestion to break the rather long paragraphs.
3. The results section could start at line 209 with the analysis of the scenarios and also Eq. 17 is as much a result as Eq. 21, this also automatically moves the only figure into the results section.
Actually, everything written until eq. (18) (plus the two paragraphs below eq. (18)) belongs to classical MFA and summarizes only known/published methods. We would therefore like keep it in the “Preliminaries” section (renamed from “Methods”; this was demanded by the Editor), while the Results section presents the new developments.
4. Line 276: It would be constructive to have an example at hand during the development of the redundancy scenario.
Actually, the scenarios S2, S3, S4, and S6 deal with redundant scenarios and how they are treated with classical MFA. Or does the reviewer mean redundant scenarios with conservation relations (CRs)? But this would require a completely new example since the one in Figure 1 does (intentionally) not contain CRs. Also, even if a network with CRs would be added, there is no specific effect that could be illustrated graphically, because the effect would be purely related to the rank of the full matrix N and the scenario-specific matrix Nu. There are also no differences in the way the scenarios are treated with CRs, the only difference is how the degrees of redundancies are calculated (and how redundancy is defined and detected). We therefore think that it would not be that useful or of much help to add another example.
Minor points:
1. Some Equations would benefit from a repetition of variable names close by: 'A' in Eq. 3, 'cT' in Eq. 4, and 'n', 'k', 'm' in line 151.
We followed the suggestion of the reviewer and added/repeated text for these variable names.
2. Line 232: R3 uses a value of 4 (not 2) for S3 in the Figure 1.
Thanks, corrected.
3. Line 394: Which number should be chosen for an infinite weight? It is counter intuitive that inactive reactions are assigned to a high weight (experimental uncertainty).
First of all, please note that a high weight (penalty) implies a very LOW uncertainty of the given/measured rate. For example, when choosing w_i = 1/(sigma_i) ^2, this means that the variance is close to zero. Since reaction rates with known zero value are often well-defined (e.g. knockout of a gene/reaction; non-availability of external substrates or of oxygen etc.), it makes sense to use very large weights for them to indicate high certainty. In fact, many solvers allow setting infinity for the weights implying that the associated flux value (here: of zero) cannot be changed. As written in the text, for reactions with fixed zero flux, this would have the same effect as removing the reaction from the network.
4. Line 541 CNapy-> CNApy
Thanks, corrected.
Reviewer 2 Report
The authors describe methods for resolving inconsistencies in constraint-based approaches, particularly those originating from the integration of flux measurements. The more simple approach for MFA, which is based on least-squares and the pseudoinverse of the stoichiometric matrix, is outlined, and the authors explain why it is not applicable to the more realistic constraint-based approaches that include reaction reversibility and other constraints. The authors then describe a similar approach for detecting and resolving the arising inconsistencies by minimizing weighted errors of measured fluxes. Hints for useful weights, based on measurement variance, are given. The methods are clearly explained and illustrated with examples. The approaches are very relevant for the practical integration of omics data with metabolic models.
From a practical perspective, it would be interesting to know about the performance of the linear and quadratic programs in typical genome-scale networks with a few thousands of reactions and varying numbers of measured fluxes. Do the corrections lead to numerical problems, or the mentioned alternative optima, in practice when applied to large networks? What could be done to evaluate alternative optima? The authors briefly discuss application to genome-scale networks. It would be ideal if the authors could present the results of the application to a realistic genome-scale scenario in more detail.
As far as I remember, in this paper:
Agren et al. PLoS Comput Biol 2012;8(5):e1002518 (10.1371/journal.pcbi.1002518)
the authors resolved infeasibilities by allowing for small deviations of metabolite concentrations from steady state. If deemed relevant, it would be interesting to discuss the similarities/differences with the here presented approach.
line 168: For completeness, consider writing out this condition.
Fig. 1: the figure shows 'I' for inconsistent, the legend uses 'IC'.
line 548: "know" -> "known"
Author Response
Reviewer 2
The authors describe methods for resolving inconsistencies in constraint-based approaches, particularly those originating from the integration of flux measurements. The more simple approach for MFA, which is based on least-squares and the pseudoinverse of the stoichiometric matrix, is outlined, and the authors explain why it is not applicable to the more realistic constraint-based approaches that include reaction reversibility and other constraints. The authors then describe a similar approach for detecting and resolving the arising inconsistencies by minimizing weighted errors of measured fluxes. Hints for useful weights, based on measurement variance, are given. The methods are clearly explained and illustrated with examples. The approaches are very relevant for the practical integration of omics data with metabolic models.
We thank the reviewer for this positive assessment.
From a practical perspective, it would be interesting to know about the performance of the linear and quadratic programs in typical genome-scale networks with a few thousands of reactions and varying numbers of measured fluxes. Do the corrections lead to numerical problems, or the mentioned alternative optima, in practice when applied to large networks? What could be done to evaluate alternative optima? The authors briefly discuss application to genome-scale networks. It would be ideal if the authors could present the results of the application to a realistic genome-scale scenario in more detail.
Following the suggestion of the reviewer, we now tested the different approaches (QP/LP with different weighting schemes) with two different solvers (CPLEX vs. MATLAB) in the core as well as in the genome-scale model via three different example scenarios of infeasibility (with increasing complexity). The results are presented at the end of the Results chapter (end of section 3.7). In summary, it turns out that the LP-based approach finds (with both solvers) in all problems in the core and in the genome-scale model very quickly a solution (less than one second). The same holds true for the QP-based solution when CPLEX’s QP solver (cplexqp) is used, while we encountered some problems with MATLAB’s QP solver (quadprog) in the genome-scale problems. Hence, at least in some cases, the LP-based approach might be more practical than the QP-based approach.
Regarding alternative optima: please note that, as also mentioned in the text, the QP-based approach has always a unique optimal solution with respect to the corrections (while the unknown reactions rates may be non-unique, as in classical FBA). For the LP-approach, we already mentioned in section 3.3., that alternative optima become unlikely in case of weights that are derived from the flux measurements (e.g. weights relative to the known fluxes).
As far as I remember, in this paper: Agren et al. PLoS Comput Biol 2012;8(5):e1002518 (10.1371/journal.pcbi.1002518) the authors resolved infeasibilities by allowing for small deviations of metabolite concentrations from steady state. If deemed relevant, it would be interesting to discuss the similarities/differences with the here presented approach.
The paper cited by the reviewer does not deal with inconsistent flux/FBA scenarios. Instead, it uses the INIT algorithm to reconstruct cell-type specific models from proteomic and metabolomics data (i.e., the latter data are used to prune the full genome-scale model to a sub-model representing the active metabolism of the respective cell type). Thus, there is no direct relationship with the methods presented in our work. However, the author is right that the cited paper relaxes the steady-state condition in order to allow net accumulation (but no net consumption) of metabolites, which ensures that net production of the latter is possible at all. As described in section 3.4 (“Allowing minimal corrections of other constraints to make infeasible FBA systems feasible”), for making an infeasible system feasible, we can use our LP/QP approach also in combination with a relaxation of the steady-state condition to allow net accumulation/consumption of metabolites to obtain a feasible system (instead of changing fixed (known) reaction rates). In fact, as also mentioned in that subsection, this is what is done by the solution obtained with the pseudo-inverse in classical MFA. However, we do not see a direct relationship with the INIT method used by Agren et al. (as the latter does not deal with infeasible systems) and we also believe that allowance of a permanent net accumulation or even a permanent net consumption (i.e. a negative balance) of a metabolite is less reasonable than assuming an error in (and thus correcting of) the measured/known fluxes.
line 168: For completeness, consider writing out this condition.
We now describe this condition more explicitly.
Fig. 1: the figure shows 'I' for inconsistent, the legend uses 'IC'.
Thanks, corrected.
line 548: "know" -> "known"
Thanks, corrected.
Reviewer 3 Report
In the manuscript titled “Analyzing and resolving infeasibility in flux balance analysis of metabolic networks, authors have described methods on how to identify, resolve and proceed with infeasibility in FBA problems. Authors have addressed a very relevant challenge in the area of constraint-based modeling and metabolic flux analysis (MFA). Applying constraints (e.g. reaction rates, growth rates, substrate uptake, etc.) most of the times may result in infeasible solution. In general, the manuscript is well written and the methods are well described for ease of implementation and wider application. There are few minor gaps that can be addressed for wider acceptance and applicability of these methods. Following are my comments:
1. Authors have provided a very useful guideline to solve infeasibilities associated with constraint-based modeling specific to MFA problems. They provide implementations in CellNetAnalyzer in both MATLAB as well as python platforms. It would be useful to provide the links (that point to either the tutorials or specific implementations) for the methods described in this manuscript. It would also be very useful to know if there are other advantages (beyond having enhanced GUI features) for the newly released CNApy pertaining to solving such infeasibility problems.
2. Authors have explained the different MFA scenarios and methods used to compute corrections using a small network. The core model example for ECC2 is also very well described. They further discussed about potential challenges and difficulties associated with the E.coli GSM iJO1366. The latest GSM for E. coli is iML1515, do the authors have similar assumptions for that GSM too? In the discussion section authors should discuss a bit about how to address such challenges associated with GSM or large scale models.
3. For wider applicability of the methods described in this work. I would like to understand how these methods perform when it comes to multi-compartment models (e.g. Eukaryotic systems) or even larger scale networks (e.g. the Recon 3D). How challenging or different would the computation be with respect to computation time solver selection, etc.
4. Authors are highly recommended to provide links to the specific implementations wherever possible all through the manuscript. If authors have designed tutorials around the specific methods described in this manuscript, they should be mentioned in the methods section.
5. Provide details about the specific solvers used or required for implementation of the different methods – the LP and QP formulations. Specifically, that would result in the reaction rates provided in Figure 1
6. In the figure description replace IC: inconsistent with I: inconsistent
Author Response
Reviewer #3:
*************
In the manuscript titled “Analyzing and resolving infeasibility in flux balance analysis of metabolic networks, authors have described methods on how to identify, resolve and proceed with infeasibility in FBA problems. Authors have addressed a very relevant challenge in the area of constraint-based modeling and metabolic flux analysis (MFA). Applying constraints (e.g. reaction rates, growth rates, substrate uptake, etc.) most of the times may result in infeasible solution. In general, the manuscript is well written and the methods are well described for ease of implementation and wider application.
We thank the reviewer for this positive assessment.
There are few minor gaps that can be addressed for wider acceptance and applicability of these methods. Following are my comments:
1. Authors have provided a very useful guideline to solve infeasibilities associated with constraint-based modeling specific to MFA problems. They provide implementations in CellNetAnalyzer in both MATLAB as well as python platforms. It would be useful to provide the links (that point to either the tutorials or specific implementations) for the methods described in this manuscript. It would also be very useful to know if there are other advantages (beyond having enhanced GUI features) for the newly released CNApy pertaining to solving such infeasibility problems.
We now provide links to the respective tools. The use of the methods is described in the respective manuals/documentation of the tools. The treatment of infeasibilities in CNA and CNApy are analogous, so there is no major difference.
2. Authors have explained the different MFA scenarios and methods used to compute corrections using a small network. The core model example for ECC2 is also very well described. They further discussed about potential challenges and difficulties associated with the E.coli GSM iJO1366. The latest GSM for E. coli is iML1515, do the authors have similar assumptions for that GSM too? In the discussion section authors should discuss a bit about how to address such challenges associated with GSM or large scale models.
We expect no major differences with the iML1515 model (in particular, the made discussion of carbon balances (or balances of other elements) will hold their as well). The reason for using the previous E. coli GSM model iJO1366 is that the ECC2 model used herein as core model was derived from the latter and so we wanted to use iJO1366 as reference model.
Actually, the section on the genome-scale model did not pose/discuss any particular challenges regarding the treatment of infeasibility (it rather discusses when infeasibility may occur), so we are not sure to what challenge the reviewer refers. However, at the end of this section, we now also provide data on several tests of the developed methods to resolve infeasibility in the small-scale and in the genome-scale model indicating that, in genome-scale models, the LP-approach may, at least in some cases (or for some solvers), be more practical than the QP-based approach.
3. For wider applicability of the methods described in this work. I would like to understand how these methods perform when it comes to multi-compartment models (e.g. Eukaryotic systems) or even larger scale networks (e.g. the Recon 3D). How challenging or different would the computation be with respect to computation time solver selection, etc.
Generally, the key indicator for the runtime is normally the size of the model (not the number of compartments). For the LP-based approach, one can expect similar runtimes as for doing FBA in those models, which appears to be well feasible with state-of-the art solvers. The QP-based approach may become too elaborate in those very large networks, at least with some solvers as indicated by the added benchmarks in section 3.7.
4. Authors are highly recommended to provide links to the specific implementations wherever possible all through the manuscript. If authors have designed tutorials around the specific methods described in this manuscript, they should be mentioned in the methods section.
We now provide links to the tools. The use of the methods is described in the respective manuals/documentation of CNA and CNApys. We do not have specific tutorials since the use be straightforward with the description in the manuals.
5. Provide details about the specific solvers used or required for implementation of the different methods – the LP and QP formulations. Specifically, that would result in the reaction rates provided in Figure 1.
We now mention explicitly the solvers supported in CNA and CNApy. Note that the QP-approach has always a unique optimal solution (as also described in the text), so there is no need to specify the solver in Figure 1 for the two scenarios (S8 and S9) that used the QP-based approach. All other scenarios (S1-S7) are treated with the pseudo-inverse resulting in unique solutions.
6. In the figure description replace IC: inconsistent with I: inconsistent
Thanks, corrected.
Round 2
Reviewer 1 Report
Thank you for considering the suggestions.